# Assessment of the Offensive Play in Elite Water Polo Using the Team Sport Assessment Procedure (TSAP) over an Entire Competitive Season

**DOI:** 10.3390/jfmk8030130

**Published:** 2023-09-05

**Authors:** Andrea Perazzetti, Milivoj Dopsaj, Mauro Mandorino, Antonio Tessitore

**Affiliations:** 1Faculty of Sport and Physical Education, University of Belgrade, 11000 Belgrade, Serbia; perazzettiandrea@gmail.com (A.P.); milivoj.dopsaj@fsfv.bg.ac.rs (M.D.); 2Department of Movement, Human and Health Sciences, University of Rome ‘Foro Italico’, 00135 Rome, Italy; mmandorino@parmacalcio1913.com; 3Performance and Analytics Department, Parma Calcio 1913, 43121 Parma, Italy

**Keywords:** match analysis, offensive phase, TSAP, decision making, final score difference, match location

## Abstract

In water polo, the team’s technical and tactical performance is determined by the sum of the players’ activities. This study aimed to investigate the playing offensive performance of an Italian First League team performed during all matches (n = 19) of the 2021/22 championship using the Team Sport Assessment Procedure (TSAP). For all subjects (n = 15), gaining possession of the ball (received balls (RB) and conquered balls (CB)) and disposing of the ball (neutral balls (NB); lost balls (LB); offensive ball (OB) and successful Shots (SS)) parameters, as well as volume of play (VP), efficiency index (EI) and performance score (PS) indexes, were analyzed in relation to the playing positions, season phase, match location and final score difference. Multiple linear regression showed a significant association between the playing position and VP and PS. Perimetral players showed the highest VP (65%) and PS (66%) values, and center defenders showed the highest values of CB (30%), while center forwards gained the highest amount of exclusion when handling the ball (48%). Although they were not significant, the other contextual factors showed that season phase and match location could affect the TSAP indexes. For water polo coaches, the TSAP represents an effective tool to assess how players interpret the match.

## 1. Introduction

Water polo is an intermittent, high-intensity and body-contact aquatic team sport [1] played by two teams of seven players (one goalkeeper and six field players) with the object to score points by throwing the ball into the opponent team’s goal [2].

The last decade has seen an increase in interest for research lines related to time motion and notational analyses [3], even though a body of water polo literature has long been focused on players’ physiological [4], anthropometric [5] and swimming profiles [6], as well as on training and match loads monitoring [1,7]. Specifically, the analysis of team differences in technical and tactical parameters in relation to various competitive levels has been conducted using a notational analysis approach [8,9,10,11,12].

Men’s national and international matches were compared in a study by Lupo et al. [13], showing that the competitive level (sub-elite and elite) had a positive impact on the occurrence of even, counterattack and power play situations and that elite teams executed more complex offensive actions than the sub-elite ones. In a study over the course of three consecutive seasons of the male Spanish Professional Water Polo League, by grouping the teams into “strong,” “medium,” and “weak” levels based on their final league table positions, Pérez et al. [14] investigated the same game situations, showing that the “strong level” teams made more counterattacks and goals scored than the “weak level” ones. The final score difference between balanced (1–3 goals) and unbalanced (>3 goals) games in elite water polo suggests that more power play actions occur in balanced games [15], whereas unbalanced games have a lower frequency of even actions and a higher number of counterattacks, indicating that defensive players may not be able to address the opponents’ counterattacks [16]. Additionally, two studies that focused on elite men’s matches in the Spanish First League championship examined the effect of match location, showing that having a home advantage, particularly in the final period, affected the result of the match [17,18].

The 2019 revision of the water polo rules of play (Federation Internationale De Natation, FINA) states that each team may retain ball possession for up to 30 s during a regular action or for up to 20 s in case of events as exclusion, corner throw or rebound to the attacking team after a shot (including after a penalty shot). Players’ individual skills, such as ball technique, accuracy when passing the ball while under pressure from the opponent and shooting accuracy, are crucial to successfully completing a ball possession phase [19]. These technical skills are also connected to players’ individual tactical knowledge, which reflects their ability to read the play situations [20] and make decisions in the most efficient and accurate manner possible.

However, despite the fact that the analysis of the number and duration of ball possession phases [21] combined with the analysis of players’ tactical decisions [22] could be helpful to aid coaching staffs in better understanding the offensive behavior of their own team, there are still only a few studies on player decision making. One of the causes of this scarcity of studies may be the lack of familiarity with useful observational instruments for water polo practitioners used to objectively evaluate players’ tactical awareness and game knowledge.

The Team Sport Assessment Procedure (TSAP) was first developed by Grehaigne et al. [23] in the contexts of pre-assessment in physical education classes, followed by ecological and concurrent validity studies [24,25,26] and focusing on teaching/learning aims [27]. Over the years, this observational instrument has been used in several team sports to evaluate individual and collective technical and tactical efficiency, analyzing how players gain and lose possession of the ball and providing specific performance indexes [28,29,30] in soccer [31,32,33], ice hockey [34,35], basketball [36,37,38] and rugby [39]. Nevertheless, to the best of our knowledge, its use in water polo studies is still restricted to a study conducted on elite young players [40], and there are no studies on elite senior players.

Therefore, the purpose of this study was to use the TSAP tool to provide coaches with objective data on individual and team offensive performances in relation to the phase of the competitive season, match location, final score difference and playing positions for male elite water polo players. The study’s hypotheses were that over the course of a competitive water polo season, there might be noticeable variations in the TSAP performance indexes between the regular phase and play-out matches, balanced and unbalanced matches, home and away matches, as well as based on the players’ playing position.

## 2. Materials and Methods

### 2.1. Experimental Design

All matches’ footage were downloaded after each weekly match (first access on 30 September 2021) from the ‘ADrive.com’ database (https://www.adrive.com (accessed on 30 September 2021)) (Emeryville, CA, USA), which is a platform with free access to clubs and where the Italian First League water polo clubs were obliged by the Italian Swimming Federation to upload a professional video (file.mp4, 1920 × 1080 16:9 HD 1080, 44 khz) of all their matches after their public streaming on YouTube or other public platforms. For this reason, since the current study used footage with free public access, no informed consent was required according to the ethical standards outlined by the local research ethics committees.

All matches were classified according to the following contextual factors: (a) season phase, based on the Italian water polo First League, which divided the 2021/22 championship schedule into regular season and play-out and play-off phases; (b) match location (home or away); (c) final score difference, which is defined as the difference of goals scored between the two teams in a match (balanced = ≤3 goals or unbalanced = >3 goals) [15]; and (d) playing time, which is defined as the player’s total minutes of play (including stoppages and interruptions during play) in each official match [1].

### 2.2. Subjects

Fifteen male elite water polo players belonging to the Italian elite First League team ‘S.S. Lazio Nuoto’ were observed during all 19 official matches of the 2021/22 Italian men’s First League championship (Serie A1) and classified according to their principal playing position as: perimetral (or peripheral) players (n = 8); center defenders (n = 3); center forwards (n = 2); and goalkeepers (n = 2) [41].

### 2.3. Data Collection

Following them being downloaded, all the matches were reproduced using QuickTime for Mac OS X (v10.7.0, Apple), which allowed the researchers to pause the video and use replay. Then, a specific water polo dashboard was created using LongoMatch Pro software (LongoMatch by Fluendo, Windows version 1.9), which allowed the researchers to collect the TSAP parameters throughout all individual and team offensive actions. In line with previous notational analysis studies on water polo [15,16,21], the same professional match analyst with extensive experience with elite water polo scored all the matches twice, showing an intraclass correlation coefficient (ICC) never below than 0.98 for each TSAP parameter.

Notational analysis data are provided by means of the TSAP observational instrument (Grehaigne et al., 1997), including received balls (RB) and conquered balls (CB) counted as variables for gaining the ball possession, while neutral balls (NB), lost balls (LB), offensive balls (OB) and successful shots (SS) were counted as variables for disposing of the ball (Table 1). As performance indicators the volume of play (VP = RB + CB), efficiency index (EI = (OB + SS)/(10 + LB)) and performance score (PS = (VP/2) + (EI*10)) were calculated from the TSAP variables [40].

### 2.4. Statistical Analysis

Statistical analyses were performed using Excel version 2016 (Microsoft Office) and SPSS version 26.0 (IBM, Chicago, IL, USA). For each TSAP variable, descriptive data are expressed as means ± standard deviations (SD).

Two multiple linear regressions were generated, with VP and PS as dependent variables, while in both analyses, the playing position, season phase, match location and final score difference were the constant predictors. Then, the analysis of covariance (ANCOVA) with playing time as a covariate was applied to assess differences in the VP and PS in relation to the playing positions. ANCOVA analysis was shown to be robust to violations of either the conditional normality or homoscedasticity assumption in previous studies [42]. For this reason, the normality assumption was not checked. Post hoc pairwise comparisons were assessed using the Bonferroni test. The results of the pairwise comparisons are described as mean difference ± standard error (SE) and *p*-value.

Finally, the k-means cluster analysis was applied to identify 3 groups of players (‘dominant’, ‘good’ and ‘less decisive’) [34] in relation to their individual VP and PS indexes. The level of significance was set at *p* < 0.05.

## 3. Results

Following the regular phase (thirteen matches), the observed team participated in the play-out phase (six matches) reserved for the last seven teams of the final league table at the end of the regular phase. Of these nineteen official matches, ten were played at home, and nine matches were played away. Regarding the final score difference factor, ten matches registered a balanced result (≤3 goals), while nine matches ended with an unbalanced result (>3 goals).

Table 2 shows the descriptive statistics (minimum, maximum, mean, SD and %CV) of the team’s TSAP parameters calculated based on the pooled data (19 matches).

Figure 1, Figure 2 and Figure 3 show the percentage of technical-tactical components for the LB, OB and SS parameters in relation to the pooled data (19 matches). Regarding the average (mean ± SD) of LB technical-tactical components, 14.2 ± 3.8 shots were registered, passages 5.8 ± 2.5, 5.7 ± 3.8 balls were stolen and 1.9 ± 1.2 contrafoul actions were performed per match. For the OB components, 13.4 ± 5.1 passages were registered, 7.7 ± 2.8 center balls were registered and gained exclusions with ball 4.3 ± 2.2 actions per match were registered, while for the SS components, 8.3 ± 2.6 scored goals were registered, 2.6 ± 2 corners were gained, 0.4 ± 0.6 shots at the end of quarter were registered and 1.0 ± 1.2 other kind of shot actions were observed per match.

Figure 4 illustrates the VP, EI and PS indexes values for each match in relation to the match location (home and away).

Multiple linear regression analyses, except those for the playing position, showed no significance for the season phase, match location and final score difference in terms of the VP and PS indexes (Table 3 and Table 4).

Table 5 shows the descriptive statistics (mean ± SD) of the team’s TSAP parameters calculated based on the pooled data (19 matches) in relation to the season phase, match location and final score difference.

Table 6 provides descriptive statistics of the VP, EI and PS indexes of all the players according to their playing position.

According to the players’ playing position, the highest VP mean values (mean ± SE) were registered for perimetral players (22.9 ± 0.8), followed by center defenders (18.8 ± 1.1), center forwards (9.8 ± 1.4) and goalkeepers (6.5 ± 1.7). ANCOVA analysis (with Bonferroni pairwise comparison) showed significant differences for the VP index between the goalkeepers and perimetral players (*p* < 0.001; mean difference ± SE = −16.3 ± 1.9), goalkeepers and center defenders (*p* < 0.001; mean difference ± SE = −12.3 ± 21), perimetral players and center defenders (*p* = 0.031; mean difference ± SE= 4 ± 1.4), perimetral players and center forwards (*p* < 0.001; mean difference ± SE = 13 ± 1.7) and between center defenders and center forwards (*p* < 0.001; mean difference ± SE = 8.9 ± 1.8). 

For PS, significant differences were found between the goalkeepers and perimetral players (*p* < 0.001; mean difference ± SE = −12.2 ± 1.2), goalkeepers and center defenders (*p* < 0.001; mean difference ± SE = 18.8 ± 1.1), goalkeepers and center forwards (*p* = 0.014; mean difference ± SE = −4.5 ± 1.4), perimetral players and center defenders (*p* = 0.001; mean difference ± SE = 3.4 ± 0.9), perimetral players and center forwards (*p* < 0.001; mean difference ± SE= 7.7 ± 1) and between the center defenders and center forwards (*p* = 0.001; mean difference ± SE = 4.2 ± 1.1). 

The k-means cluster analysis identified three groups according to the VP and PS indexes (reported as mean and SD). The first cluster included eight players (VP = 7.2 ± 3.7; PS = 4.9 ± 2.5), and the second cluster included five players (VP = 22.4 ± 6; PS = 14.1 ± 3.4), while the third cluster included only two players (VP = 38.2 ± 3.5; 24.8 ± 2). Figure 5 illustrates the individual bidimensional representation of the three clusters.

## 4. Discussion

This study aimed to use the TSAP tool to provide water polo coaches objective data on individual and team offensive performances in relation to the phase of the 2021/22 competitive season (regular and play-out phases), match location (home or away), final score difference (balanced = ≤3 goals or unbalanced = >3 goals) and playing positions among male elite (Italian Serie A1 championship) water polo players.

The main findings of this study were that the TSAP parameters registered different values for the four playing positions (perimetral, center defender, center forward, and goalkeeper) and that cluster analysis provided three groups of players for the VP and PS indexes.

The analysis of technical-tactical components of the TSAP parameters showed, as the main results, that 51% of the LB was determined by the shots, 53% of the OB was determined by the assist and offensive passages, while 68% of the SS was determined by scoring a goal. In a water polo match, the LB, OB and SS parameters should be considered as intertwined since their interactions could provide a positive or a negative outcome for the team. The high number of missed shots registered as LB could be explained by the players’ technical and decision-making skills. Indeed, a missed shot could lead a team to concede a counterattack or a goal to their opponents, as previously revealed in a study by Perazzetti et al. [43] on elite youth national teams, which showed a very strong negative correlation (r = −0.892) between LB and the goals scored.

Based on our study’s hypotheses, that there might be a noticeable variation in the TSAP indexes between the regular phase and play-out matches, balanced and unbalanced matches, and home and away matches; analysis of these three contextual factors and the players’ playing positions was conducted. Multiple linear regression analysis showed a not significant relationship between the season phase, match location and final score difference and the VP and PS indexes, while only the players’ playing positions showed a significant adherence.

Although they were not statistically significant, our study found that the VP and PS indexes were higher during the play-out phase, which is chronologically played after the regular one. Even though this is the first study investigating the season phase factor in water polo, the results are consistent with earlier research on other elite invasion team sports like football and Australian football, which showed that the technical and tactical performances improved toward the last part of the competitive season [44,45]. An additional explanation could be that during the six play-out matches, the observed team played against opponent teams with a lower defensive play level compared to the opponent teams faced during the regular phase.

Considering the pooled data for the entire season (regular and play-out phases), our results showed higher mean values for the balanced matches compared to those of the unbalanced matches for the PS (138 ± 15 vs. 134 ± 23, respectively). This result could be explained by the higher percentage of goals scored the during power play situation in balanced matches than that in unbalanced ones (40% vs. 29%, respectively). In fact, Lupo et al. [15] showed that, in these types of matches, the opponent team is more likely to commit exclusion fouls to execute power play actions that could result in more goals being scored, while Medić et al. [46] suggested that scoring in situations where one team has a numerical advantage over the other (power play goals) is one of the key technical and tactical factors to take into account in balanced matches.

Additionally, in our study, the analysis of the match location factor revealed higher mean values in the TSAP parameters and indexes (Table 5) for home matches (even considering the unbalanced matches and the stronger opponent teams), which is consistent with a study by Ruano et al. [18] that showed a negative impact on the teams playing away. In fact, apart the presence of an audience with local supporters and the lack of traveling, the larger influence on the TSAP values is given by the fact that, in home matches, a team plays in the same water polo swimming pool where they perform their daily training. Furthermore, in our study, the home matches were played in a historical Olympic swimming pool (the ‘Mosaic pool of Foro Italico’) which has a smaller width dimension (18 m) and different measures of depth between the two halves of the pool; therefore, these characteristics could have caused the observed team to play in a more efficiency way at home than they did in the away fields.

Regarding the playing position variable, ANCOVA analysis showed significant differences between the different positional roles, with perimetral players registering the highest mean values for both the VP and PS indexes. Although the goalkeeper position is typically excluded from water polo studies [47], we made the decision to include it in light of the interesting results we observed. In fact, according to the findings of our study, the two players who alternately occupies the goalkeeper position actively contributed to 7.7% and 6.5% of the team’s total amount of the VP and PS, respectively. The fact that, nowadays, goalkeepers collaborate with teammates in collective actions much more than they did in the past has also been shown in other invasion team sports, such as football [48,49]. For this reason, the goalkeeper’s ability to be involved in building the offensive phase of play starting from their passing the ball should be more considered by water polo coaches. The 2019 change in the water polo rules (FINA, 2019) also favored this tactical behavior by allowing the goalkeeper to pass beyond and touch the ball past the half-way line, increasing their ability to perform OB and SS actions in addition to NB actions. The highest values for the VP and PS indexes shown by perimetral players could be affected by the number of minutes of play. In our analysis, the playing time covariate could have influenced the differences registered between the playing positions considering the higher amount of minutes of play per match [1,50], as well as the larger involvement in offensive phases during play registered by the perimetral players compared to that of the center positions (i.e., CD and CF) [8]. Consequently, the perimetral players’ performances could have explained the highest values of the VP and PS for this playing position. This fact allowed the researchers to identify perimetral players as the most important ones for disposing of ball possession in water polo, while center defenders and center forwards could be considered as the main playing positions involved in fighting and wrestling situations with opponents [51]. In fact, usually, center defenders are engaged in a higher number of ‘*duel*’ actions before performing the offensive phase. As demonstrated in the study by Özkol et al. [52], the high number of defensive activities by the center defenders limits their possibility of supporting their teammates during the offensive phase. For instance, in our study, the center defenders performed 30% of the total CB for the team, demonstrating a better defensive approach compared to that of the other playing positions. On the other hand, considering the dyadic relationship between a player and their opponent, the center forward is very often engaged in a duel with his/her direct opponent to obtain the best position on the field to receive the ball. In fact, the role of a center forward is more inclined to the finalization of the offensive phase than the construction of the play [53], as showed in a study by Lupo et al. [54] on elite men’s water polo matches (n = 11), in which most of exclusion fouls (around 68%) where performed by center forwards. This fact is in line with our study, where most of the instances of exclusion by handling the ball (48.1%) were performed by those in this playing position.

The k-means cluster analysis provided in this study on the pooled data identified three groups of players based on their values of the VP and PS. According to the classification suggested by Nadeau et al. [34], in our study, the three groups were named as ‘dominant players’ (n = 2), ‘good players’ (n = 5) and ‘less-decisive players’ (n = 8) in relation to their ability to manage ball possession through offensive actions. Based on this classification, the team’s captain (player 5) and the player with the most experience (player 8) were deemed to be the “dominant players” group, while four perimetral players and one center defender were considered for the ‘good players’ group. Finally, two center defenders, two center forwards, two goalkeepers and two more perimetral players were included in the ‘less-decisive players’ group.

This kind of analysis, using such a classification, could be helpful for water polo head coaches as a source of valuable information to organize players’ substitutions during matches, plan specific trainings drills, as well as to be used at the beginning of the season for the roster construction, since many head coaches still base their choices on the players’ defensive skills rather than their offensive ones [55].

## 5. Conclusions

Based on the observation of all official matches played during the 2021/22 Italian First League water polo championship by a male elite team, through technical and tactical analyses of the offensive actions, this study has provided sufficient information on elite players’ offensive skills, as well as their ability to dispose of the ball. However, over an entire season, this research included only one elite men senior team, so future research might include a larger number of teams, as well as different competitive levels (elite and non-elite), age categories (senior and different youth categories) and different gender competitions (men and women).

The Team Sport Assessment Procedure was initially designed by Grehaigne et al. [23] for both the scientific and teaching fields. Despite its limit of obtaining information only about the phase of playing with the ball (i.e., gaining and disposing of ball possession), over the years, it has been used to a greater extent in competitive sports also during games. However, live observation could present difficulties for the observers (i.e., player identification, angle of the camera, etc.) that might result in coding errors; this is why experienced and trained observers (match analyst) are required. However, coding errors decreased when one expert observer (as was the case for our study) watches the same team more than once.

The use of TSAP as an observational instrument has confirmed that it is a valid tool to assess players’ playing abilities, which is why the coaching staffs could regularly adopt it to monitor training sessions (i.e., during small-sided games) and competitions, as an important and useful tool for monitoring players.

However, even if it is restricted to the analysis of only offensive games, this research can be regarded as an original and beneficial contribution to a better comprehension of water polo performances. Furthermore, considering that the TSAP observational tool was designed to assess only players’ offensive behavior, to provide a more complete technical and tactical analysis, this tool should also include their defensive behavior. For this reason, it could be interesting to elaborate and validate specific water polo defense indexes to correlate them with the TSAP ones.

## Figures and Tables

**Figure 1 jfmk-08-00130-f001:**
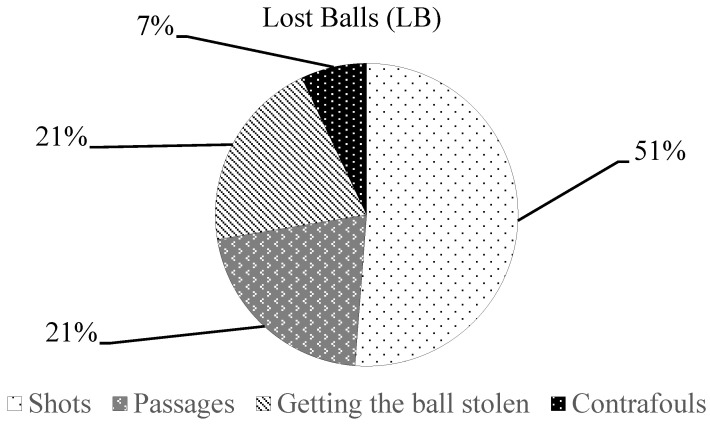
Percentage distribution of technical-tactical components of LB parameter.

**Figure 2 jfmk-08-00130-f002:**
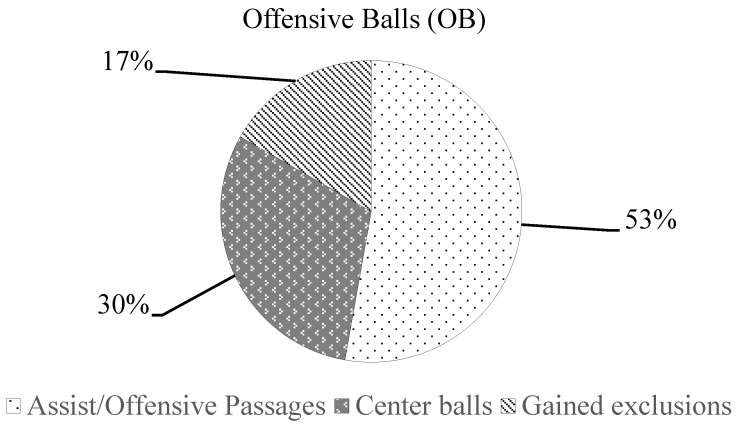
Percentage distribution of technical-tactical components of OB parameter.

**Figure 3 jfmk-08-00130-f003:**
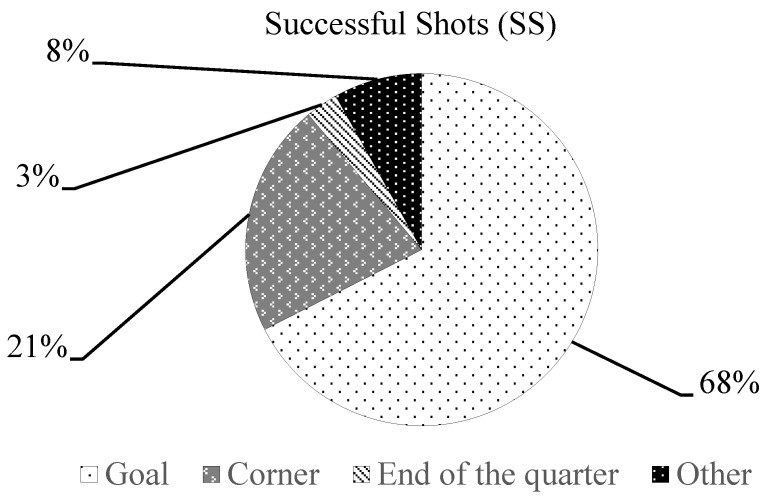
Percentage distribution of technical-tactical components of SS parameter.

**Figure 4 jfmk-08-00130-f004:**
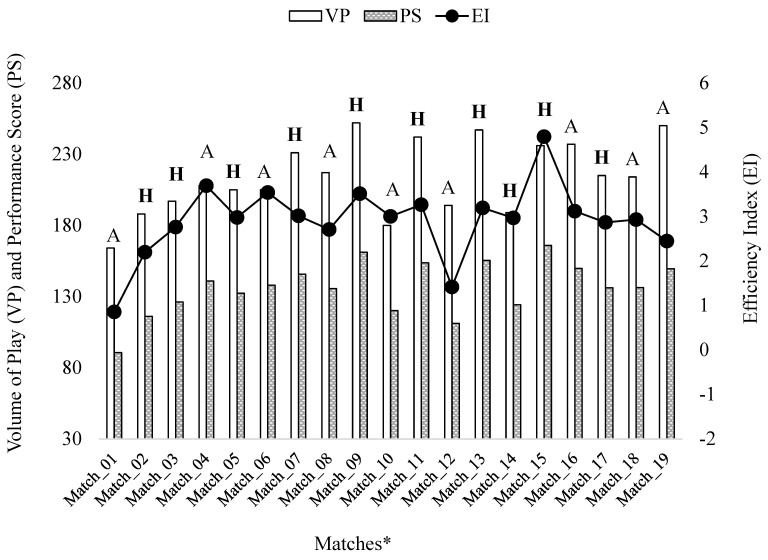
Distribution of VP, EI and PS values of each match played both away and at home location. * Note. A = away; **H** = home.

**Figure 5 jfmk-08-00130-f005:**
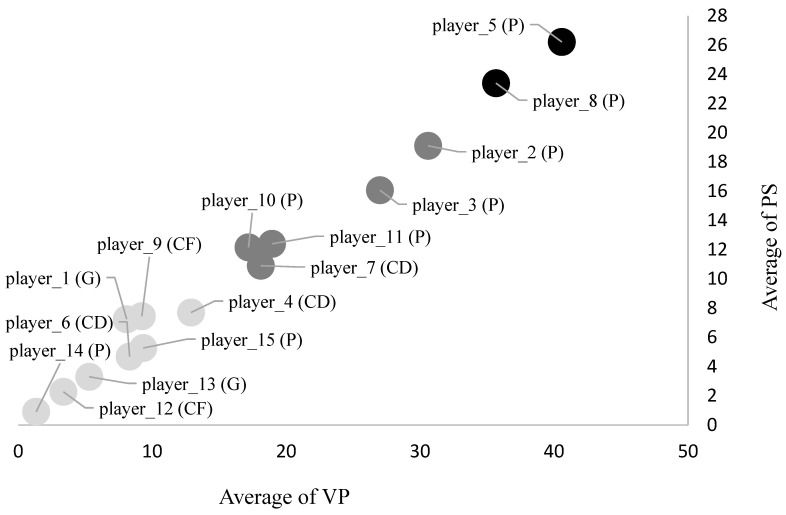
The k-means cluster analysis identifying three groups of players according to the two indexes of VP and PS. Note. VP = volume of play; PS = performance score.

**Table 1 jfmk-08-00130-t001:** TSAP Components in water polo * [40].

**GAINING POSSESSION OF THE BALL**	
Receiving the ball (RB)	A player receives the ball from a teammate without losing control of it.
Conquering the ball (CB)	A player is considered to have conquered a ball if he or she intercepts it, steals it from an opponent or recaptures it after an unsuccessful shot on goal or after a near-loss to the other team.
**DISPOSING OF THE BALL**	
Playing a neutral ball (NB)	A routine pass to a teammate or any pass which does not truly put the opponent team in jeopardy is considered a neutral ball.
Losing the ball (LB)	A player is considered as having lost the ball when he or she loses it to the other team without having scored a goal (i.e., shot: a missing shot; passage: a passage intercepted by an opponent or a failed passage; lost ball: a ball stolen by an opponent; contrafoul: a foul committed while holding the ball).
Playing an offensive ball (OB)	An offensive ball is a pass to a teammate which puts pressure on the other team and, most often, leads to a shot toward the goal (i.e., assist/offensive passage: a passage executed before a shot; center Ball: a passage toward the center forward) or a gained exclusion (i.e., while holding the ball, gaining a power play situation through an exclusion foul committed by an opponent).
Executing a successful shot (SS)	A shot is considered successful when it scores, or the possession of the ball is retained by one’s team (Goal and Shots)
**PERFORMANCE INDICATORS**	
Volume of play (VP)	The volume of play represents the number of times the player has gained possession of the ball, as calculated using the formula: VP: RB + CB
Efficiency index (EI)	The efficiency index indicates the players’ efficiency in ball possession according to their disposing of the ball, as calculated using the formula: EI: (OB + SS)/(10 + LB)
Performance score (PS)	The performance score indicates the players’ ratio between the number of times players gain the possession of the ball and their efficiency in disposing of it, as calculated using the formula: PS: (VP/2) + (EI*10)

* Adapted from Grehaigne et al. [23].

**Table 2 jfmk-08-00130-t002:** Descriptive statistics (minimum, maximum, mean, SD and %CV) of the team’s TSAP parameters with pooled data.

	MIN	MAX	MEAN	SD	%CV
RB	160	247	207.1	26.3	12.7
CB	4	16	7.2	3.2	44.7
VP	164	252	214.3	25.6	11.9
NB	115	188	149.1	21.0	14.1
LB	19	36	27.4	4.4	16.2
OB	8	38	25.4	7.3	28.8
SS	5	20	12.3	3.4	27.9
EI	0.9	4.8	2.9	0.8	28.6
PS	90.5	165.9	136.2	18.7	13.7

Note. RB = received balls; CB = conquered balls; VP = volume of play; NB = neutral balls; LB = lost balls; OB = offensive balls; SS = successful shots; EI = efficiency index; PS = performance score.

**Table 3 jfmk-08-00130-t003:** Multiple linear regression output for VP index.

	Unstandardized Coefficients		Standardized Coefficients	t	Sig.
	B	Std. Error	Beta		
(constant)	23.63	5.23		4.520	0.00
season phase	1.90	2.05	0.07	0.924	0.36
match location	−0.51	1.74	−0.02	−0.294	0.77
final score difference	0.03	1.92	0.00	0.015	0.99
**playing position**	**−5.08**	**0.97**	**−0.33**	**−5.258**	**0.00**

**Table 4 jfmk-08-00130-t004:** Multiple linear regression output for PS index.

	Unstandardized Coefficients		Standardized Coefficients	t	Sig.
	B	Std. Error	Beta		
(constant)	15.81	3.44		4.597	0.00
season phase	1.12	1.35	0.06	0.825	0.41
match location	−0.54	1.15	−0.03	−0.473	0.64
final score difference	−0.55	1.26	−0.03	−0.434	0.67
**playing position**	**−2.90**	**0.64**	**−0.29**	**−4.559**	**0.00**

**Table 5 jfmk-08-00130-t005:** Descriptive statistics (mean ± SD) of the team’s TSAP parameters with pooled data in relation to the contextual factors (season phase, match location and final score).

	Competition Phase		Match Location		Final Score Difference	
	Regular Season (13 Matches)	Play-Out (6 Matches)	Home (10 Matches)	Away (9 Matches)	Balanced (10 Matches)	Unbalanced (9 Matches)
**RB**	203 ± 28	215 ± 24	212 ± 27	202 ± 27	204 ± 22	210 ± 31
**CB**	7 ± 2	9 ± 5	9 ± 4	6 ± 1	8 ± 3	6 ± 3
**VP**	210 ± 27	224 ± 22	220 ± 24	208 ± 27	212 ± 22	216 ± 30
**NB**	145 ± 22	157 ± 18	154 ± 20	144 ± 22	145 ± 18	154 ± 24
**LB**	28 ± 4	26 ± 6	26 ± 4	30 ± 4	26 ± 4	29 ± 4
**OB**	25 ± 8	27 ± 7	27 ± 6	24 ± 9	29 ± 5	22 ± 8
**SS**	12 ± 3	13 ± 4	14 ± 3	10 ± 3	13 ± 3	11 ± 4
**EI**	2.8 ± 0.8	3.2 ± 0.8	3.2 ± 0.7	2.6 ± 0.9	3.2 ± 0.6	2.6 ± 0.9
**PS**	132.8 ± 19.9	143.6 ± 14.6	141.6 ± 17	130.2 ± 19.5	138.4 ± 14.6	133.8 ± 23.1

Note. RB = received balls; CB = conquered balls; VP = volume of play; NB = neutral balls; LB = lost balls; OB = offensive balls; SS = successful shots; EI = efficiency index; PS = performance score.

**Table 6 jfmk-08-00130-t006:** Descriptive statistics (mean ± SD) according to players’ playing position (PP).

Player	PP	N_Matches	Playing Time (Min)		TSAP Indexes		
		Total	Total	Average Per Match	VP	EI	PS
player_1	G	17	783	46.1 ± 10.1	8.1 ± 4.4	0.04 ± 0.09	7.2 ± 2.5
player_2	P	18	645	35.8 ± 7.9	30.6 ± 9.8	0.39 ± 0.24	19.1 ± 6.6
player_3	P	6	197	32.8 ± 8.2	27 ± 14.4	0.25 ± 0.24	16.1 ± 9.1
player_4	CD	17	398	23.4 ± 8.5	12.9 ± 7.4	0.13 ± 0.15	7.7 ± 4.9
player_5	P	19	788	41.5 ± 7.9	40.6 ± 12.2	0.59 ± 0.29	26.2 ± 8.1
player_6	CD	19	299	15.7 ± 6.2	8.3 ± 5.5	0.05 ± 0.06	4.7 ± 2.9
player_7	CD	19	561	29.6 ± 7.6	18.1 ± 8.2	0.19 ± 0.11	10.9 ± 4.6
player_8	P	18	783	43.5 ± 6.9	35.7 ± 4.3	0.55 ± 0.27	23.4 ± 3.8
player_9	CF	18	605	33.6 ± 5.9	9.2 ± 2.1	0.28 ± 0.15	7.4 ± 2.2
player_10	P	16	578	36.1 ± 8.6	17.2 ± 8.6	0.36 ± 0.23	12.2 ± 6.1
player_11	P	19	649	34.2 ± 8.2	18.9 ± 6.5	0.29 ± 0.14	12.4 ± 3.9
player_12	CF	18	304	16.9 ± 8.4	3.4 ± 2.1	0.06 ± 0.09	2.3 ± 1.6
player_13	G	9	224	24.9 ± 17.2	5.3 ± 5.4	0.02 ± 0.04	3.3 ± 3.1
player_14	P	12	70	5.8 ± 4.7	1.3 ± 1.3	0.02 ± 0.04	0.9 ± 1
player_15	P	3	73	24.3 ± 6.4	9.3 ± 1.15	0.06 ± 0.05	5.3 ± 0.6

Note. G = goalkeepers; P = perimetral player; CD = center defenders; CF = center forwards; VP = volume of play; EI = efficiency index; PS = performance score.

## Data Availability

Not applicable.

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
