# Peer review of "Assessment of the Offensive Play in Elite Water Polo Using the Team Sport Assessment Procedure (TSAP) over an Entire Competitive Season"

_jfmk, 2023, doi:10.3390/jfmk8030130_

Round 1

Reviewer 1 Report

Dear author,
The use of TSAPs to detect water polo team tactics is an interesting topic. However, additions are still needed to refine the visualization of the manuscript. Therefore, the following suggestions are put forward, and the author is expected to make improvements.
1. Introduction
It is recommended that the authors describe or cite other relevant sports using TSAP detection indicators in this article.
2. Materials and Methods
Like the previous question. Although the researchers put forward the detection indicators in Table 1, they found that they only detect the offensive indicators. Does this manuscript only consider offensive tactics? Does it not consider discussing defensive tactics?
If this is the guess, please explain and clarify it in the text.
3. Results
Table 3, please define the significant index of t value. p<0.05? P<0.01?
4. Discussion
Although the analysis and discussion are quite detailed, if you can make suggestions for improving the number of battles, it will add points to the manuscript.
For example, the highest values ​​of VP and PS indices exhibited by peripheral players may be affected by game time. So how to make good use of it or how to improve it?
5. Conclusions
The conclusion should be based on the author's personal opinion on the research findings, and it is not suitable to put literature to support the conclusion.
I look forward to correcting the manuscript,

Author Response

Dear reiewers,

We appreciated your time and effort dedicated to providing feedback to our manuscript and we are grateful for the insightful comments on and valuable improvements to our paper. We have followed all your suggestions and changes are available in “word” review mode in the full manuscript.

Based on your suggestions, we revised the full manuscript, in red you will find all changes.

Reviewer 1

Introduction  

Reviewer 1 comment: “It is recommended that the authors describe or cite other relevant sports using TSAP detection indicators in this article”.

Authors answer: Line 77-78. We have cited studies on invasion team sports. Specifically, soccer, ice-hockey, basketball, and rugby

Materials and Method

Reviewer 1 comment: “Like the previous question. Although the researchers put forward the detection indicators in Table 1, they found that they only detect the offensive indicators. Does this manuscript only consider offensive tactics? Does it not consider discussing defensive tactics? If this is the guess, please explain and clarify it in the text”.

Authors answer: In the Introduction and in the method sessions we carefully described that the TSAP method is referred only to the ball possession phase (offensive phase). In fact, the TSAP observational instrument is composed by two variables that describe how players gain the ball and other four variables that indicate how players manage the ball.

Results

Reviewer 1 comment: “Table 3, please define the significant index of t value. p<0.05? P<0.01?”

Authors answer: We have reproduced the SPSS table, as also suggested di statisticians.

 Discussion

Reviewer 1 comment: “Although the analysis and discussion are quite detailed, if you can make suggestions for improving the number of battles, it will add points to the manuscript.
For example, the highest values ​​of VP and PS indices exhibited by peripheral players may be affected by game time. So how to make good use of it or how to improve it?”.

Authors answer: Following the reviewer 1 suggestions, we rephrased and improved the applications of our results.

  1. Conclusions

Reviewer 1 comment: “The conclusion should be based on the author's personal opinion on the research findings, and it is not suitable to put literature to support the conclusion.
I look forward to correcting the manuscript”.

Authors answer: Following the reviewer 1 suggestion, the reference “Perazzetti, A.; Dopsaj, M.; Nedeljković, A.; Mazić, S.; Tessitore, A. Survey on coaching philosophies and training methodologies of water polo head coaches from three different European national schools. Kinesiology. 2023, 55, 49-61.” and the related sentence has been moved from  the conclusion to the discussion.

Reviewer 2 Report

Title: Assessing Offensive Phase in Elite Water Polo: The Use of Team Sport Assessment Procedure (TSAP)

Major concerns:

-            Firstly, the English throughout the manuscript need to be improved. There are many awkward sentences – and whilst I do understand what the authors are trying to say, it makes the reading very difficult and very interruptive. Please, please, and please, get someone to read and improve the sentences structure of your manuscript. For examples, Line 39, Line 62 [“most right”?], Line 83-84, Line 87 [“to”?], Line 119 [“proper”?]; Line 243 [“not”]; Line 274-278 awkward sentence. Ine 327 [“most expertise player”?], and many other sentences….

-            The purpose of the study is not clear. The TSAP is reliable and valid measure but was performed in other sports. Now the authors are trying to do the same in water polo – I get this. But to show reliability, the authors need to analyse the same matches twice and to report reliability of the analyses – this was not done. As reflected by Line 121 to 123, “From the co-authors, a professional water polo match analysts (AP) has observed twice all matches”. Clearly, more details of this “professional analyst is required”. For example, how many years of has this person have in this skill of analysing water polo performance, what is his background, etc. But the most important point that was missing from this current manuscript is the reliability of the analysis made by this “professional water polo match analyst“. Since it was stated that the individual viewed all the matches twice, then clearly there should be a measure reported of the reliability of his analyses from these two viewings – but alas no reliability data was reported in the present manuscript.

-            In regard to validity, there is a need to compare the current performance analysis of the water polo players result with a ‘gold-standard” of performance analyses (i.e., as an example from a video notional analysis) in order to say that the performance analyses measured using TSAP is indeed “accurate”. This again was not done. Authors merely reported the performance analyses taken from TSAP – without making any comparison of the analyses to something. Thus, we can’t be sure that the data collected using TSAP is actually accurately when we have nothing to compared it against!

-            In short, the authors need to verify that the data taken from TSAP is firstly reliable, and secondly validated against another gold standard measure of match performance analysis. Just because TSAP is reliable and valid in football and other team sports – it does not mean it is directly reliable and valid in water polo as well – there is a need to provide the necessary proof and evidence for this, and only then the TSAP can be deemed as useful to the water polo coach or player or performance analysts – who are going to use it.

-            Line 242 – 244. It was stated, “The multiple linear regression analysis showed a not significant relationship between season phase, match location, and final score difference, and the VP and PS indexes. Despite this result, the descriptive statistics of the two indexes provides different trends for each contextual factor.” – If there is no significant difference, it means there is no difference between the variables – a ‘trend’ is only when the p value is very close to 0.05!

-            Line 265 to 277. I don’t understand the purpose of this paragraph since it already bene shown there is no significant difference in location of matches. Secondly, the number of home and away matches were ‘even’, 6 vs 6 – hence I cannot comprehend how come the authors claim that home matches were the key for the TSAP values because the away matches will also tend to lower the TSAP values in return.

-            I am also concern of the use of abstracts as references. Reference 36 and Ref 50.  To my knowledge abstract is not acceptable as valid references.

Minor issues:

-            The aim of the study and the title of the study does not tally. While the title seems to seek the usefulness of the TSAP – but the conclusion seems to highlight the fact that the TSAP is a valid measure of performance or match analysis of water polo.

-            Throughout the manuscript. I think the term, “play-out” is not appropriate. I think a better term is “play-off”. So please change.

-            Line 33. “along the years” is not a good term to use.

-            Line 41. What is “man-up”? You mean a power-play with an extra man advantage because the opponent has one-man suspended.

-            Line 42. “elaborate”?

-            Line 45. final league table position.  

-            Line 47-49. Why 3 goals are set as the threshold for scores differences. What is the rationale or justification for the use of 3 goals? Is there a published study to support this?

-            Line 107. Does the total minutes of play include stoppages and interruptions during play. Please define accurately this term.

-            Line 149. What are the “3 groups of players”? please clarify.

-            For the TSAP components in Table 1. What are the unit of measures for each of the variable /component? Is it frequency or number of times or per min or what? Need to inform the reader. Secondly for VP, EI and PS – there is a need to inform the reader what does the term implies. For example the EI – efficient in what sense? In short, what does the VP, EI and PS scores actually means – as an example, the authors should define these terms and explain the scores in relation to water polo. There is a need to provide perspectives to the three terms to the reader. In addition, in Line 223. What do the terms “ two indexes of VP and PS”?

-            Line 169 and 172. What are “passages”, “contrafouls”, “assist/offensive passages”, “center balls” and “gained exclusions”? . These terms are not common and have not clearly defined and are now straightaway being used! The authors must keep in mind that the readers/reviewers are not water polo experts and thus it is the responsibility of the authors to ‘educate” them and not assume they know these terms.

-            What is “dyadic”? is that some sports relationship?

please above

Author Response

Dear reiewers,

We appreciated your time and effort dedicated to providing feedback to our manuscript and we are grateful for the insightful comments on and valuable improvements to our paper. We have followed all your suggestions and changes are available in “word” review mode in the full manuscript.

Based on your suggestions, we revised the full manuscript, in red you will find all changes.

Reviewer 2

Major concerns

Reviewer 2 comment: “Firstly, the English throughout the manuscript need to be improved. There are many awkward sentences – and whilst I do understand what the authors are trying to say, it makes the reading very difficult and very interruptive. Please, please, and please, get someone to read and improve the sentences structure of your manuscript. For examples, Line 39, Line 62 [“most right”?], Line 83-84, Line 87 [“to”?], Line 119 [“proper”?]; Line 243 [“not”]; Line 274-278 awkward sentence. Ine 327 [“most expertise player”?], and many other sentences….”.

Authors answer: We revised and modified the full manuscript improving the English language style.

Reviewer 2 comment: “The purpose of the study is not clear. The TSAP is reliable and valid measure but was performed in other sports. Now the authors are trying to do the same in water polo – I get this. But to show reliability, the authors need to analyse the same matches twice and to report reliability of the analyses – this was not done. As reflected by Line 121 to 123, “From the co-authors, a professional water polo match analysts (AP) has observed twice all matches”. Clearly, more details of this “professional analyst is required”. For example, how many years of has this person have in this skill of analysing water polo performance, what is his background, etc. But the most important point that was missing from this current manuscript is the reliability of the analysis made by this “professional water polo match analyst“. Since it was stated that the individual viewed all the matches twice, then clearly there should be a measure reported of the reliability of his analyses from these two viewings – but alas no reliability data was reported in the present manuscript”.

Authors answer: We reported the reliability data in the manuscript in term of ICC, after we performed the intraclass correlation analysis.          
The author that was officially employed as a first match analyst of the team in the season 2021-22, in that period had more than 100 analysed matches for senior teams (A1 and A2 leagues) and more than 250 matches for elite young categories.

Reviewer 2 comment: “In regard to validity, there is a need to compare the current performance analysis of the water polo players result with a ‘gold-standard” of performance analyses (i.e., as an example from a video notional analysis) in order to say that the performance analyses measured using TSAP is indeed “accurate”. This again was not done. Authors merely reported the performance analyses taken from TSAP – without making any comparison of the analyses to something. Thus, we can’t be sure that the data collected using TSAP is actually accurately when we have nothing to compared it against!”

Authors answer: TSAP is a unique match analysis tool that take in consideration only the offensive phase emphasizing each event according to the possession of the ball (how each player got the ball and what each player does with the ball). For this reason, is not possible to compare with traditional match analysis usually performed by water polo practitioners. We consider this fact a useful novelty for this aquatic discipline.

Reviewer 2 comment: “In short, the authors need to verify that the data taken from TSAP is firstly reliable, and secondly validated against another gold standard measure of match performance analysis. Just because TSAP is reliable and valid in football and other team sports – it does not mean it is directly reliable and valid in water polo as well – there is a need to provide the necessary proof and evidence for this, and only then the TSAP can be deemed as useful to the water polo coach or player or performance analysts – who are going to use it”.

Authors answer:  It is not the first paper published in water polo. Perazzetti et al. (2023) demonstrated that in the U20 World championship, the winner team (Greece) registered the highest values of TSAP parameters and indices compared to the teams that got the second, third and fourth place. Now in this study, we cannot compare with other gold standard measures, since we would like to give information about the same team during an entire competitive season.

Reviewer 2 comment: “Line 242 – 244. It was stated, “The multiple linear regression analysis showed a not significant relationship between season phase, match location, and final score difference, and the VP and PS indexes. Despite this result, the descriptive statistics of the two indexes provides different trends for each contextual factor.” – If there is no significant difference, it means there is no difference between the variables – a ‘trend’ is only when the p value is very close to 0.05!”

Authors answer: Yes, it is true but since this is TSAP is quite a novelty in the field of water polo, we would like to discuss the divergent values provided by the descriptive statistics (even if no significant) to give a general idea of the use of this instrument according to the contextual factors that we have put in consideration.

Reviewer 2 comment:  “Line 265 to 277. I don’t understand the purpose of this paragraph since it already bene shown there is no significant difference in location of matches. Secondly, the number of home and away matches were ‘even’, 6 vs 6 – hence I cannot comprehend how come the authors claim that home matches were the key for the TSAP values because the away matches will also tend to lower the TSAP values in return”.

Authors answer: We have rephrased all the sentence.

Reviewer 2 comment: “I am also concern of the use of abstracts as references. Reference 36 and Ref 50.  To my knowledge abstract is not acceptable as valid references”.

Authors answer: In previous manuscripts published by our group, also in JFMK, the reference of an abstract has been accepted. We agree with reviewer 2 that ìf there is enough literature abstracts should not be used as references. This was the case only due to the scarcity of literature on the topic.

Minor issues

Reviewer 2 comment: “The aim of the study and the title of the study does not tally. While the title seems to seek the usefulness of the TSAP – but the conclusion seems to highlight the fact that the TSAP is a valid measure of performance or match analysis of water polo”.

Authors answer:  The TSAP is only a valid measure for the ball possession.

Reviewer 2 comment: “Throughout the manuscript. I think the term, “play-out” is not appropriate. I think a better term is “play-off”. So please change”.

Authors answer: In team sport, play-off to refer to a series of matches between teams on the top of the same round table or different round tables, to decide which teams will play for winning a championship, while play-outs are matches played at the end of a season between teams with placements that would lead to relegation. Since the analysed team, after the regular phase, competed in a group of other 6 teams for the relegation, we used the term play-out.

Reviewer 2 comment:  “Line 33. “along the years” is not a good term to use”.

Authors answer:  Line 76. We followed your suggestion, and we used the term “Over the years” instead of “Along the yers”.

Reviewer 2 comment: “Line 41. What is “man-up”? You mean a power-play with an extra man advantage because the opponent has one-man suspended”.

Authors answer:  Line 41 and line 48. We followed your suggestion and we used the term “power-play” instead of “man-up” term.

Reviewer 2 comment: “Line 42. “elaborate”?”

Authors answer: we changed in “complex”

Reviewer 2 comment: “Line 45. final league table position”.  

Authors answer: We added the term “league”

Reviewer 2 comment: “Line 47-49. Why 3 goals are set as the threshold for scores differences. What is the rationale or justification for the use of 3 goals? Is there a published study to support this?”

Authors answer:  It is a parameter used in team sports to assess the “intensity” of the match. We added the reference  Lupo, C., Condello, G., & Tessitore, A. (2012). Notational analysis of elite men’s water polo related to specific margins of victory. Journal of sports science & medicine, 11(3), 516.

Reviewer 2 comment:  “Line 107. Does the total minutes of play include stoppages and interruptions during play. Please define accurately this term”.

Authors answer: Playing time has been defined as the player’s total minutes of play in each official match, which also included the player exclusion during a ‘man-down’ situation.

Reviewer 2 comment: “Line 149. What are the “3 groups of players”? please clarify”.

Authors answer: Following reviewer 2 suggestions we have clarified it

Reviewer 2 comment: “For the TSAP components in Table 1. What are the unit of measures for each of the variable /component? Is it frequency or number of times or per min or what? Need to inform the reader. Secondly for VP, EI and PS – there is a need to inform the reader what does the term implies. For example the EI – efficient in what sense? In short, what does the VP, EI and PS scores actually means – as an example, the authors should define these terms and explain the scores in relation to water polo. There is a need to provide perspectives to the three terms to the reader. In addition, in Line 223. What do the terms “ two indexes of VP and PS”?”.

Authors answer:  It is number of occurance during the match.

VP is the total number of time that a player have the possession of the ball and it is composed by RB+CB.

EI is the efficiency index composed by the formula (OB+SS)/(10+LB) and indicate the efficiency of each player during the ball possession according to their disposing of the ball. In particular the division is composed by the sum of Offensive balls and successful shot divided 10 plus the lost ball. The higher value of this index means a good number of OB and SS compared to the LB showing a player that positively influences the offensive phase of the team. Furthermore this index is necessary for the calculation of PS.

PS is the performance score composed by the formula (VP/2)+(EI*10). Higher is this index it means that a player disposing very well the number of balls that received or conquered.

Reviewer 2 comment: Line 169 and 172. What are “passages”, “contrafouls”, “assist/offensive passages”, “center balls” and “gained exclusions”? . These terms are not common and have not clearly defined and are now straightaway being used! The authors must keep in mind that the readers/reviewers are not water polo experts and thus it is the responsibility of the authors to ‘educate” them and not assume they know these terms.

Authors answer: We followed your suggestion and we improved the explanations of all terms included in the Table 1.

GAINING POSSESSION OF THE BALL

Receiving the ball
(RB)

A player receives the ball from a teammate without losing the control of it.

Conquering the ball
(CB)

A player is considered having conquered a ball if he or she intercepts it, stoles it from an opponent, or recaptures it after an unsuccessful shot on goal or after a near-loss to the other team.

DISPOSING OF THE BALL

Playing a neutral ball
(NB)

A routine pass to a teammate or any pass which does not truly put the opponent team in jeopardy is considered a neutral ball.

Losing the ball
(LB)

A player is considered having lost the ball when he or she loses it to the other team without having scored a goal (i.e., Shot: a missing shot; Passage:  a passage intercepted by an opponent or a failed passage; Lost Ball: a ball stolen by an opponent; Contrafoul: a foul committed while holding the ball).

Playing an offensive ball (OB)

An offensive ball is a pass to a teammate which puts pressure on the other team and, most often, leads to a shot on goal (i.e., Assist/Offensive passage: a passage executed before a shot: Center Ball: a passage toward the center forward) or a gained exclusion (i.e., while holding the ball, gaining a power-play situation through an exclusion foul committed by an opponent).

Executing a successful shot (SS)

    A shot is considered successful when it scores, or possession of the ball is retained by one's team (Goal and Shots)

PERFORMANCE INDICATORS

Volume of Play
(VP)

The volume of play represents the number of times the player has gained possession of the ball, calculated by the formula:

VP: RB+CB

Efficiency Index
(EI)

The efficiency index indicates the players’ efficiency in ball possession according to their disposing of the ball, calculated by the formula:

EI: (OB+SS)/(10+LB)

Performance Score
(PS)

The performance score indicates the players’ ratio between the number of times players gain the possession of the ball and their efficiency in disposing it, calculated by the formula:

PS: (VP/2)+(EI*10)

Reviewer 2 comment: What is “dyadic”? is that some sports relationship?

Authors answer:  Dyadic is a sport term used to describe a group of two opponents. In this case was for give a more open explanation of the confrontation of two players that is not only composed by a fighting duel.

Round 2

Reviewer 1 Report

Dear author,
It is a pleasure to receive this manuscript, I think the authors have responded well to questions and suggestions and this manuscript can proceed to the next stage of the process.
good luck,

Author Response

On behalf of the authors we deeply thank reviewer 1 for his/her comment and good wishes

Reviewer 2 Report

please include the limitations of the tsap 

na

Author Response

On behalf of the authors we deeply thank reviewer 2 for his/her comment and requirement which will improve the manuscript

The following sentence (written in red colour and highlighted in yellow) has been added from line 342 to line 349

The Team Sport Assessment Procedure was designed by Grehaigne et al. [23] for both scientific and teaching fields. Despite its limit on obtaining information only about the phase of playing with the ball (i.e., gaining and disposing of ball possession) over the years it has seen a greater extent in competitive sports, also during the game. However, live observation could present difficulties for the observers (i.e., player identification, angle of the camera, etc.) that might result in coding errors, reason why experienced and trained observer (match analyst) are required. However, coding errors decreased when one expert observer (as it has been for our study) watches the same team more than once.

Round 3

Reviewer 2 Report

okay